# Systematic Generalization: What Is Required and Can It Be Learned?

**Dzmitry Bahdanau**[*]
Mila, Université de Montréal
AdeptMind Scholar
Element AI

**Shikhar Murty**[*]
Mila, Université de Montréal

**Michael Noukhovitch**
Mila, Université de Montréal

**Thien Huu Nguyen**
University of Oregon

**Harm de Vries**
Mila, Université de Montréal

**Aaron Courville**
Mila, Université de Montréal
CIFAR Fellow

## Abstract

Numerous models for grounded language understanding have been recently proposed, including (i) generic models that can be easily adapted to any given task and (ii) intuitively appealing modular models that require background knowledge to be instantiated. We compare both types of models in how much they lend themselves to a particular form of systematic generalization. Using a synthetic VQA test, we evaluate which models are capable of reasoning about all possible object pairs after training on only a small subset of them. Our findings show that the generalization of modular models is much more systematic and that it is highly sensitive to the module layout, i.e. to how exactly the modules are connected. We furthermore investigate if modular models that generalize well could be made more end-to-end by learning their layout and parametrization. We find that end-to-end methods from prior work often learn inappropriate layouts or parametrizations that do not facilitate systematic generalization. Our results suggest that, in addition to modularity, systematic generalization in language understanding may require explicit regularizers or priors.

## 1 Introduction

In recent years, neural network based models have become the workhorse of natural language understanding and generation. They empower industrial machine translation (Wu et al., 2016) and text generation (Kannan et al., 2016) systems and show state-of-the-art performance on numerous benchmarks including Recognizing Textual Entailment (Gong et al., 2017), Visual Question Answering (Jiang et al., 2018), and Reading Comprehension (Wang et al., 2018). Despite these successes, a growing body of literature suggests that these approaches do not generalize outside of the specific distributions on which they are trained, something that is necessary for a language understanding system to be widely deployed in the real world. Investigations on the three aforementioned tasks have shown that neural models easily latch onto statistical regularities which are omnipresent in existing datasets (Agrawal et al., 2016; Gururangan et al., 2018; Jia & Liang, 2017) and extremely hard to avoid in large scale data collection. Having learned such dataset-specific solutions, neural networks fail to make correct predictions for examples that are even slightly out of domain, yet are trivial for humans. These findings have been corroborated by a recent investigation on a synthetic instruction-following task (Lake & Baroni, 2018), in which seq2seq models (Sutskever et al., 2014; Bahdanau et al., 2015) have shown little systematicity (Fodor & Pylyshyn, 1988) in how they generalize, that is they do not learn general rules on how to compose words and fail spectacularly when for example asked to interpret "jump twice" after training on "jump", "run twice" and "walk twice".

An appealing direction to improve the generalization capabilities of neural models is to add modularity and structure to their design to make them structurally resemble the kind of rules they are

---

[*]Equal contribution

supposed to learn (Andreas et al., 2016; Gaunt et al., 2016). For example, in the Neural Module Network paradigm (NMN, Andreas et al. (2016)), a neural network is assembled from several *neural modules*, where each module is meant to perform a particular subtask of the input processing, much like a computer program composed of functions. The NMN approach is intuitively appealing but its widespread adoption has been hindered by the large amount of domain knowledge that is required to decide (Andreas et al., 2016) or predict (Johnson et al., 2017; Hu et al., 2017) how the modules should be created (*parametrization*) and how they should be connected (*layout*) based on a natural language utterance. Besides, their performance has often been matched by more traditional neural models, such as FiLM (Perez et al., 2017), Relations Networks (Santoro et al., 2017), and MAC networks (Hudson & Manning, 2018). Lastly, generalization properties of NMNs, to the best of our knowledge, have not been rigorously studied prior to this work.

Here, we investigate the impact of explicit modularity and structure on systematic generalization of NMNs and contrast their generalization abilities to those of generic models. For this case study, we focus on the task of visual question answering (VQA), in particular its simplest binary form, when the answer is either "yes" or "no". Such a binary VQA task can be seen as a fundamental task of language understanding, as it requires one to evaluate the truth value of the utterance with respect to the state of the world. Among many systematic generalization requirements that are desirable for a VQA model, we choose the following basic one: *a good model should be able to reason about all possible object combinations despite being trained on a very small subset of them*. We believe that this is a key prerequisite to using VQA models in the real world, because they should be robust at handling unlikely combinations of objects. We implement our generalization demands in the form of a new synthetic dataset, called **S**patial **Q**ueries **O**n **O**bject **P**airs (SQOOP), in which a model has to perform spatial relational reasoning about pairs of randomly scattered letters and digits in the image (e.g. answering the question "*Is there a letter A left of a letter B?*"). The main challenge in SQOOP is that models are evaluated on all possible object pairs, but trained on only a subset of them.

Our first finding is that NMNs do generalize better than other neural models when layout and parametrization are chosen appropriately. We then investigate which factors contribute to improved generalization performance and find that using a layout that matches the task (i.e. a tree layout, as opposed to a chain layout), is crucial for solving the hardest version of our dataset. Lastly, and perhaps most importantly, we experiment with existing methods for making NMNs more end-to-end by inducing the module layout (Johnson et al., 2017) or learning module parametrization through soft-attention over the question (Hu et al., 2017). Our experiments show that such end-to-end approaches often fail by not converging to tree layouts or by learning a blurred parameterization for modules, which results in poor generalization on the hardest version of our dataset. We believe that our findings challenge the intuition of researchers in the field and provide a foundation for improving systematic generalization of neural approaches to language understanding.

## 2 THE SQOOP DATASET FOR TESTING SYSTEMATIC GENERALIZATION

We perform all experiments of this study on the SQOOP dataset. SQOOP is a minimalistic VQA task that is designed to test the model's ability to interpret unseen combinations of known relation and object words. Clearly, given known objects X, Y and a known relation R, a human can easily verify whether or not the objects X and Y are in relation R. Some instances of such queries are common in daily life (*is there a cup on the table*), some are extremely rare (*is there a violin under the car*), and some are unlikely but have similar, more likely counter-parts (*is there grass on the frisbee* vs *is there a frisbee on the grass*). Still, a person can easily answer these questions by understanding them as just the composition of the three separate concepts. Such compositional reasoning skills are clearly required for language understanding models, and SQOOP is explicitly designed to test for them.

Concretely speaking, SQOOP requires observing a $64 \times 64$ RGB image x and answering a yes-no question $q = \mathrm{X\,R\,Y}$ about whether objects X and Y are in a spatial relation R. The questions are represented in a redundancy-free $\mathrm{X\,R\,Y}$ form; we did not aim to make the questions look like natural language. Each image contains 5 randomly chosen and randomly positioned objects. There are 36 objects: the latin letters A-Z and digits 0-9, and there are 4 relations: LEFT_OF, RIGHT_OF, ABOVE, and BELOW. This results in $36 \cdot 35 \cdot 4 = 5040$ possible unique questions (we do not allow questions about identical objects). To make negative examples challenging, we ensure that both X and Y of a question are always present in the associated image and that there are distractor objects $\mathrm{Y}' \neq \mathrm{Y}$

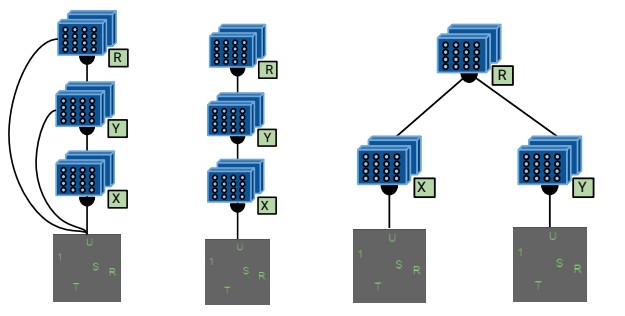
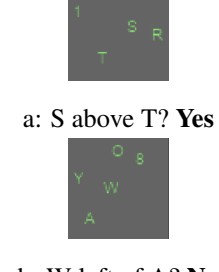

Figure 1: Different NMN layouts: *NMN-Chain-Shortcut* (**left**), *NMN-Chain* (**center**), *NMN-Tree* (**right**). See Section 3.2 for details.

Figure 2: A positive (**top**) and negative (**bottom**) example from the SQOOP dataset.

a: S above T? **Yes**

b: W left of A? **No**

and $X' \neq X$ such that $X R Y'$ and $X' R Y$ are both true for the image. These extra precautions guarantee that answering a question requires the model to locate all possible X and Y then check if any pair of them are in the relation R. Two SQOOP examples are shown in Figure 2.

Our goal is to discover which models can correctly answer questions about all $36 \cdot 35$ possible object pairs in SQOOP after having been trained on only a subset. For this purpose we build training sets containing $36 \cdot 4 \cdot k$ unique questions by sampling $k$ different right-hand-side (RHS) objects $Y_1$, $Y_2$, ..., $Y_k$ for each left-hand-side (LHS) object X. We use this procedure instead of just uniformly sampling object pairs in order to ensure that each object appears in at least one training question, thereby keeping the all versions of the dataset solvable. We will refer to $k$ as the *#rhs/lhs parameter* of the dataset. Our test set is composed from the remaining $36 \cdot 4 \cdot (35 - k)$ questions. We generate training and test sets for rhs/lhs values of 1,2,4,8 and 18, as well as a control version of the dataset, #rhs/lhs=35, in which both the training and the test set contain all the questions (with different images). Note that lower #rhs/lhs versions are harder for generalization due to the presence of spurious dependencies between the words X and Y to which the models may adapt. In order to exclude a possible compounding factor of overfitting on the training images, all our training sets contain 1 million examples, so for a dataset with #rhs/lhs = $k$ we generate approximately $10^6/(36 \cdot 4 \cdot k)$ different images per unique question. Appendix D contains pseudocode for SQOOP generation.

## 3 MODELS

A great variety of VQA models have been recently proposed in the literature, among which we can distinguish two trends. Some of the recently proposed models, such as FiLM (Perez et al., 2017) and Relation Networks (RelNet, Santoro et al. (2017)) are highly *generic* and do not require any task-specific knowledge to be applied on a new dataset. On the opposite end of the spectrum are *modular* and *structured* models, typically flavours of Neural Module Networks (Andreas et al., 2016), that do require some knowledge about the task at hand to be instantiated. Here, we evaluate systematic generalization of several state-of-the-art models in both families. In all models, the image x is first fed through a CNN based network, that we refer to as the *stem*, to produce a feature-level 3D tensor $h_x$. This is passed through a model-specific computation conditioned on the question $q$, to produce a joint representation $h_{qx}$. Lastly, this representation is fed into a fully-connected *classifier* network to produce logits for prediction. Therefore, the main difference between the models we consider is how the computation $h_{qx} = model(h_x, q)$ is performed.

### 3.1 GENERIC MODELS

We consider four generic models in this paper: CNN+LSTM, FiLM, Relation Network (RelNet), and Memory-Attention-Control (MAC) network. For CNN+LSTM, FiLM, and RelNet models, the question $q$ is first encoded into a fixed-size representation $h_q$ using a unidirectional LSTM network. **CNN+LSTM** flattens the 3D tensor $h_x$ to a vector and concatenates it with $h_q$ to produce $h_{qx}$:

$$h_{qx} = [flatten(h_x); h_q]. \tag{1}$$

**RelNet** (Santoro et al., 2017) uses a network $g$ which is applied to all pairs of feature columns of $h_{\mathrm{x}}$ concatenated with the question representation $h_q$, all of which is then pooled to obtain $h_{q\mathrm{x}}$:

$$h_{q\mathrm{x}} = \sum_{i,j} g(h_{\mathrm{x}}(i), h_{\mathrm{x}}(j), h_q) \tag{2}$$

where $h_x(i)$ is the $i$-th feature column of $h_x$. **FiLM** networks (Perez et al., 2017) use $N$ convolutional FiLM blocks applied to $h_{\mathrm{x}}$. A FiLM block is a residual block (He et al., 2016) in which a feature-wise affine transformation (FiLM layer) is inserted after the 2nd convolutional layer. The FiLM layer is conditioned on the question at hand via prediction of the scaling and shifting parameters $\gamma_n$ and $\beta_n$:

$$[\gamma_n; \beta_n] = W_q^n h_q + b_q^n \tag{3}$$

$$\tilde{h}_{q\mathrm{x}}^n = BN(W_2^n * ReLU(W_1^n * h_{q\mathrm{x}}^{n-1} + b_n)) \tag{4}$$

$$h_{q\mathrm{x}}^n = h_{q\mathrm{x}}^{n-1} + ReLU(\gamma_n \odot \tilde{h}_{q\mathrm{x}}^n \oplus \beta_n) \tag{5}$$

where $BN$ stands for batch normalization (Ioffe & Szegedy, 2015), $*$ stands for convolution and $\odot$ stands for element-wise multiplications. $h_{q\mathrm{x}}^n$ is the output of the $n$-th FiLM block and $h_{q\mathrm{x}}^0 = h_{\mathrm{x}}$. The output of the last FiLM block $h_{q\mathrm{x}}^N$ undergoes an extra $1 \times 1$ convolution and max-pooling to produce $h_{q\mathrm{x}}$. **MAC** network of Hudson & Manning (2018) produces $h_{q\mathrm{x}}$ by repeatedly applying a Memory-Attention-Composition (MAC) cell that is conditioned on the question through an attention mechanism. The MAC model is too complex to be fully described here and we refer the reader to the original paper for details.

## 3.2 NEURAL MODULE NETWORKS

Neural Module Networks (NMN) (Andreas et al., 2016) are an elegant approach to question answering that constructs a question-specific network by composing together trainable neural modules, drawing inspiration from symbolic approaches to question answering (Malinowski & Fritz, 2014). To answer a question with an NMN, one first constructs the computation graph by making the following decisions: (a) how many modules and of which types will be used, (b) how will the modules be connected to each other, and (c) how are these modules parametrized based on the question. We refer to the aspects (a) and (b) of the computation graph as the *layout* and the aspect (c) as the *parametrization*. In the original NMN and in many follow-up works, different module types are used to perform very different computations, e.g. the `Find` module from Hu et al. (2017) performs trainable convolutions on the input attention map, whereas the `And` module from the same paper computes an element-wise maximum for two input attention maps. In this work, we follow the trend of using more homogeneous modules started by Johnson et al. (2017), who use only two types of modules: unary and binary, both performing similar computations. We restrict our study to NMNs with homogeneous modules because they require less prior knowledge to be instantiated and because they performed well in our preliminary experiments despite their relative simplicity. We go one step further than Johnson et al. (2017) and retain a single binary module type, using a zero tensor for the second input when only one input is available. Additionally, we choose to use exactly three modules, which simplifies the layout decision to just determining how the modules are connected. Our preliminary experiments have shown that, even after these simplifications, NMNs are far ahead of other models in terms of generalization.

In the original NMN, the layout and parametrization were set in an ad-hoc manner for each question by analyzing a dependency parse. In the follow-up works (Johnson et al., 2017; Hu et al., 2017), these aspects of the computation are predicted by learnable mechanisms with the goal of reducing the amount of background knowledge required to apply the NMN approach to a new task. We experiment with the End-to-End NMN (N2NMN) (Hu et al., 2017) paradigm from this family, which predicts the layout with a seq2seq model (Sutskever et al., 2014) and computes the parametrization of the modules using a soft attention mechanism. Since all the questions in SQOOP have the same structure, we do not employ a seq2seq model but instead have a trainable layout variable and trainable attention variables for each module.

Formally, our NMN is constructed by repeatedly applying a *generic neural module* $f(\theta, \gamma, s^0, s^1)$, which takes as inputs the shared parameters $\theta$, the question-specific parametrization $\gamma$ and the left-hand side and right-hand side inputs $s^0$ and $s^1$. Three such modules are connected and conditioned

on a question $q = (q_1, q_2, q_3)$ as follows:

$$\gamma_k = \sum_{i=1}^{3} \alpha^{k,i} e(q_i) \tag{6}$$

$$s_k^m = \sum_{j=-1}^{k-1} \tau_m^{k,j} s_j \tag{7}$$

$$s_k = f(\theta, \gamma_k, s_k^0, s_k^1) \tag{8}$$

$$h_{qx} = s_3 \tag{9}$$

In the equations above, $s_{-1} = 0$ is the zero tensor input, $s_0 = h_x$ are the image features outputted by the stem, $e$ is the embedding table for question words. $k \in \{1, 2, 3\}$ is the module number, $s_k$ is the output of the $k$-th module and $s_k^m$ are its left ($m = 0$) and right ($m = 1$) inputs. We refer to $A = (\alpha^{k,i})$ and $T = (\tau_m^{k,j})$ as the *parametrization attention matrix* and the *layout tensor* respectively.

We experiment with two choices for the NMN's generic neural module: the Find module from Hu et al. (2017) and the Residual module from Johnson et al. (2017). The equations for the Residual module are as follows:

$$[W_1^k; b_1^k; W_2^k; b_2^k; W_3^k; b_3^k] = \gamma_k \tag{10}$$

$$\tilde{s}_k = ReLU(W_3^k * [s_k^0; s_k^1] + b_3^k), \tag{11}$$

$$f_{Residual}(\gamma_k, s_k^0, s_k^1) = ReLU(\tilde{s}_k + W_1^k * ReLU(W_2^k * \tilde{s}_k + b_2^k)) + b_1^k), \tag{12}$$

and for Find module as follows:

$$[W_1; b_1; W_2; b_2] = \theta, \tag{13}$$

$$f_{Find}(\theta, \gamma_k, s_k^0, s_k^1) = ReLU(W_1 * \gamma_k \odot ReLU(W_2 * [s_k^0; s_k^1] + b_2) + b_1). \tag{14}$$

In the formulas above all $W$'s stand for convolution weights, and all $b$'s are biases. Equations 10 and 13 should be understood as taking vectors $\gamma_k$ and $\theta$ respectively and chunking them into weights and biases. The main difference between Residual and Find is that in Residual all parameters depend on the questions words (hence $\theta$ is omitted from the signature of $f_{Residual}$), where as in Find convolutional weights are the same for all questions, and only the element-wise multipliers $\gamma_k$ vary based on the question. We note that the specific Find module we use in this work is slightly different from the one used in (Hu et al., 2017) in that it outputs a feature tensor, not just an attention map. This change was required in order to connect multiple Find modules in the same way as we connect multiple residual ones.

Based on the generic NMN model described above, we experiment with several specific architectures that differ in the way the modules are connected and parametrized (see Figure 1). In **NMN-Chain** the modules form a sequential chain. Modules 1, 2 and 3 are parametrized based on the first object word, second object word and the relation word respectively, which is achieved by setting the attention maps $\alpha_1, \alpha_2, \alpha_3$ to the corresponding one-hot vectors. We also experiment with giving the image features $h_x$ as the right-hand side input to all 3 modules and call the resulting model **NMN-Chain-Shortcut**. **NMN-Tree** is similar to NMN-Chain in that the attention vectors are similarly hard-coded, but we change the connectivity between the modules to be tree-like. **Stochastic N2NMN** follows the N2NMN approach by Hu et al. (2017) for inducing layout. We treat the layout $T$ as a stochastic latent variable. $T$ is allowed to take two values: $T_{tree}$ as in NMN-Tree, and $T_{chain}$ as in NMN-Chain. We calculate the output probabilities by marginalizing out the layout i.e. probability of answer being "yes" is computed as $p(\text{yes}|x, q) = \sum_{T \in \{T_{tree}, T_{chain}\}} p(\text{yes}|T, x, q)p(T)$. Lastly, **Attention N2NMN** uses the N2NMN method for learning parametrization (Hu et al., 2017). It is structured just like NMN-Tree but has $\alpha^k$ computed as $\text{softmax}(\tilde{\alpha}^k)$, where $\tilde{\alpha}^k$ is a trainable vector. We use Attention N2NMN only with the Find module because using it with the Residual module would involve a highly non-standard interpolation between convolutional weights.

## 4 EXPERIMENTS

In our experiments we aimed to: (a) understand which models are capable of exhibiting systematic generalization as required by SQOOP, and (b) understand whether it is possible to induce, in an end-to-end way, the successful architectural decisions that lead to systematic generalization.

All models share the same stem architecture which consists of 6 layers of convolution (8 for Relation Networks), batch normalization and max pooling. The input to the stem is a $64 \times 64 \times 3$ image, and the feature dimension used throughout the stem is 64. Further details can be found in Appendix A. The code for all experiments is available online[1].

### 4.1 WHICH MODELS GENERALIZE BETTER?

We report the performance for all models on datasets of varying difficulty in Figure 3. Our first observation is that the modular and tree-structured NMN-Tree model exhibits strong systematic generalization. Both versions of this model, with $\mathrm{Residual}$ and $\mathrm{Find}$ modules, robustly solve all versions of our dataset, including the most challenging #rhs/lhs=1 split.

The results of NMN-Tree should be contrasted with those of generic models. 2 out of 4 models (Conv+LSTM and RelNet) are not able to learn to answer all SQOOP questions, no matter how easy the split was (for high #rhs/lhs Conv+LSTM overfitted and RelNet did not train). The results of other two models, MAC and FiLM, are similar. Both models are clearly able to solve the SQOOP task, as suggested by their almost perfect $< 1\%$ error rate on the control #rhs/lhs=35 split, yet they struggle to generalize on splits with lower #rhs/lhs. In particular, we observe $13.67 \pm 9.97\%$ errors for MAC and a $34.73 \pm 4.61\%$ errors for FiLM on the hardest #rhs/lhs=1 split. For the splits of intermediate difficulty we saw the error rates of both models decreasing as we increased the #rhs/lhs ratio from 2 to 18. Interestingly, even with 18 #rhs/lhs some MAC and FiLM runs result in a test error rate of $\sim 2\%$. Given the simplicity and minimalism of SQOOP questions, we believe that these results should be considered a failure to pass the SQOOP test for both MAC and FiLM. That said, we note a difference in how exactly FiLM and MAC fail on #rhs/lhs=1: in several runs (3 out of 15) MAC exhibits a strong generalization performance ($\sim 0.5\%$ error rate), whereas in all runs of FiLM the error rate is about $30\%$. We examine the successful MAC models and find that they converge to a successful setting of the control attention weights, where specific MAC units consistently attend to the right questions words. In particular, MAC models that generalize strongly for each question seem to have a unit focusing strongly on $X$ and a unit focusing strongly on $Y$ (see Appendix B for more details). As MAC was the strongest competitor of NMN-Tree across generic models, we perform an ablation study for this model, in which we vary the number of modules and hidden units, as well as experiment with weight decay. These modifications do not result in any significant reduction of the gap between MAC and NMN-Tree. Interestingly, we find that using the default high number of MAC units, namely 12, is helpful, possibly because it increases the likelihood that at least one unit converges to focus on X and Y words (see Appendix B for details).

### 4.2 WHAT IS ESSENTIAL TO STRONG GENERALIZATION OF NMN?

The superior generalization of NMN-Tree raises the following question: what is the key architectural difference between NMN-Tree and generic models that explains the performance gap between them? We consider two candidate explanations. First, the NMN-Tree model differs from the generic models in that it does not use a language encoder and is instead built from modules that are parametrized by question words directly. Second, NMN-Tree is structured in a particular way, with the idea that modules 1 and 2 may learn to locate objects and module 3 can learn to reason about object locations independently of their identities. To understand which of the two differences is responsible for the superior generalization, we compare the performance of the NMN-Tree, NMN-Chain and NMN-Chain-Shortcut models (see Figure 1). These 3 versions of NMN are similar in that none of them are using a language encoder, but they differ in how the modules are connected. The results in Figure 3 show that for both $\mathrm{Find}$ and $\mathrm{Residual}$ module architectures, using a tree layout is absolutely crucial (and sufficient) for generalization, meaning that the generalization gap between NMN-Tree and generic models can not be explained merely by the language encoding step in the latter. In particular, NMN-Chain models perform barely above random chance, doing even worse than generic models on

---
[1]https://github.com/rizar/systematic-generalization-sqoop

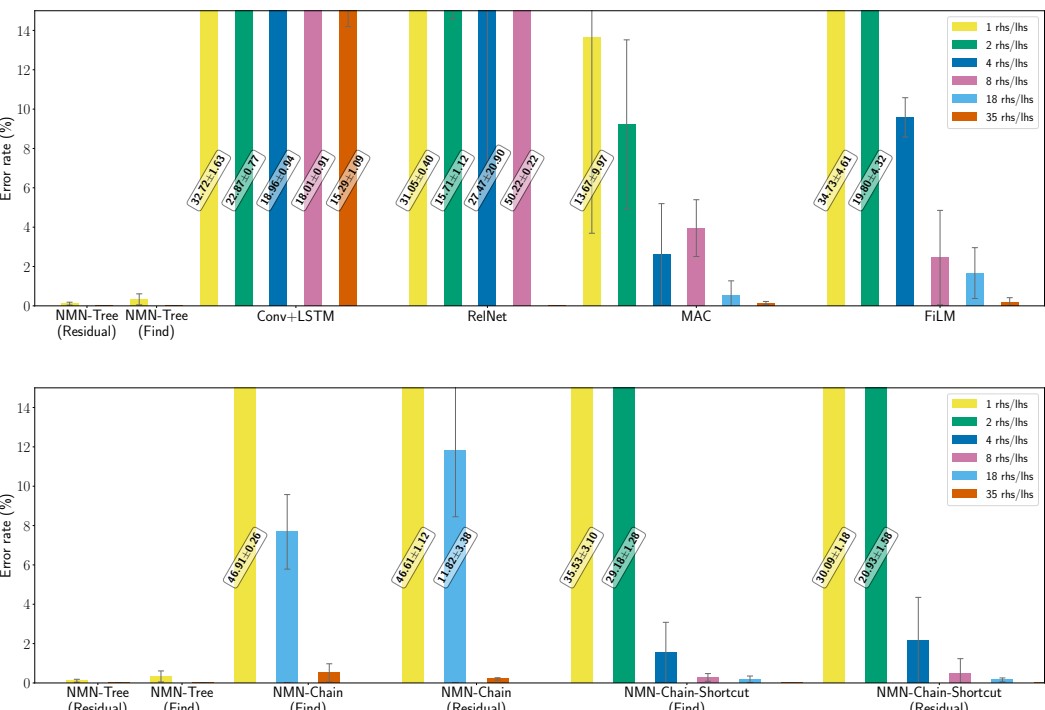

Figure 3: **Top:** Comparing the performance of generic models on datasets of varying difficulty (lower #rhs/lhs is more difficult). Note that NMN-Tree generalizes perfectly on the hardest #rhs/lhs=1 version of SQOOP, whereas MAC and FiLM fail to solve completely even the easiest #rhs/lhs=18 version. **Bottom:** Comparing NMNs with different layouts and modules. We can clearly observe the superior generalization of NMN-Tree, poor generalization of NMN-Chain and mediocre generalization of NMN-Chain-Shortcut. Means and standard deviations after at least 5 runs are reported.

the #rhs/lhs=1 version of the dataset and dramatically failing even on the easiest #rhs/lhs=18 split. This is in stark contrast with NMN-Tree models that exhibits nearly perfect performance on the hardest #rhs/lhs=1 split. As a sanity check we train NMN-Chain models on the vanilla #rhs/lhs=35 split. We find that NMN-Chain has little difficulty learning to answer SQOOP questions when it sees all of them at training time, even though it previously shows poor generalization when testing on unseen examples. Interestingly, NMN-Chain-Shortcut performs much better than NMN-Chain and quite similarly to generic models. We find it remarkable that such a slight change in the model layout as adding shortcut connections from image features $h_x$ to the modules results in a drastic change in generalization performance. In an attempt to understand why NMN-Chain generalizes so poorly we compare the test set responses of the 5 NMN-Chain models trained on #rhs/lhs=1 split. Notably, there was very little agreement between predictions of these 5 runs (Fleiss $\kappa = 0.05$), suggesting that NMN-Chain performs rather randomly outside of the training set.

## 4.3 Can the Right Kind of NMN Be Induced?

The strong generalization of the NMN-Tree is impressive, but a significant amount of prior knowledge about the task was required to come up with the successful *layout* and *parametrization* used in this model. We therefore investigate whether the amount of such prior knowledge can be reduced by fixing one of these structural aspects and inducing the other.

### 4.3.1 Layout Induction

In our layout induction experiments, we use the Stochastic N2NMN model which treats the layout as a stochastic latent variable with two values ($T_{tree}$ and $T_{chain}$, see Section 3.2 for details). We experiment with N2NMNs using both Find and Residual modules and report results with different

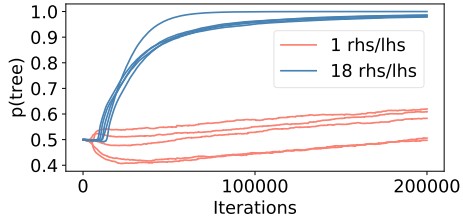
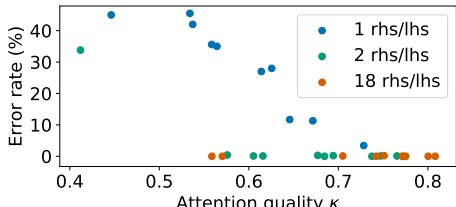

Figure 4: Learning dynamics of layout induction on 1 rhs/lhs and 18 rhs/lhs datasets using the Residual module with $p_0(tree) = 0.5$. All 5 runs do not learn to use the tree layout for 1 rhs/lhs, the very setting where the tree layout is necessary for generalization.

Figure 5: Attention quality $\kappa$ vs accuracy for Attention N2NMN models trained on different #rhs/lhs splits. We can observe that generalization is strongly associated with high $\kappa$ for #rhs/lhs=1, while for splits with 2 and 18 rhs/lhs blurry attention may be sufficient.

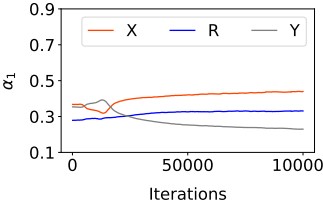
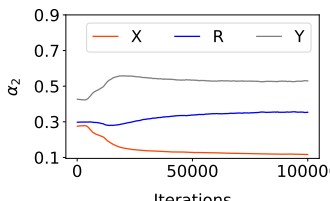
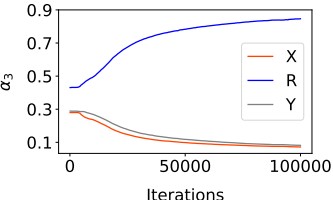

Figure 6: An example of how attention weights of modules 1 (**left**), 2 (**middle**), and 3 (**right**) evolve during training of an Attention N2NMN model on the 18 rhs/lhs version of SQOOP. Modules 1 and 2 learn to focus on different objects words, X and Y respectively in this example, but they also assign high weight to the relation word R. Module 3 learns to focus exclusively on R.

initial conditions, $p_0(tree) \in 0.1, 0.5, 0.9$. We believe that the initial probability $p_0(tree) = 0.1$ should not be considered small, since in more challenging datasets the space of layouts would be exponentially large, and sampling the right layout in 10% of all cases should be considered a very lucky initialization. We repeat all experiments on #rhs/lhs=1 and on #rhs/lhs=18 splits, the former to study generalization, and the latter to control whether the failures on #rhs/lhs=1 are caused specifically by the difficulty of this split. The results (see Table 1) show that the success of layout induction (i.e. converging to a $p(tree)$ close to $0.9$) depends in a complex way on all the factors that we considered in our experiments. The initialization has the most influence: models initialized with $p_0(tree) = 0.1$ typically do not converge to a tree (exception being experiments with Residual module on #rhs/lhs=18, in which 3 out of 5 runs converged to a solution with a high $p(tree)$). Likewise, models initialized with $p_0(tree) = 0.9$ always stay in a regime with a high $p(tree)$. In the intermediate setting of $p_0(tree) = 0.5$ we observe differences in behaviors for Residual and Find modules. In particular, N2NMN based on Residual modules stays spurious with $p(tree) = 0.5 \pm 0.08$ when #rhs/lhs=1, whereas N2NMN based on Find modules always converges to a tree.

One counterintuitive result in Table 1 is that for the Stochastic N2NMNs with Residual modules, trained with $p_0(tree) = 0.5$ and #rhs/lhs=1, make just $1.64 \pm 1.79\%$ test error despite never resolving the layout uncertainty through training ($p_{200K}(tree) = 0.56 \pm 0.06$). We offer an investigation of this result in Appendix C.

### 4.3.2 PARAMETRIZATION INDUCTION

Next, we experiment with the Attention N2NMN model (see Section 3.2) in which the parametrization is learned for each module as an attention-weighted average of word embeddings. In these experiments, we fix the layout to be tree-like and sample the pre-softmax attention weights $\tilde{\alpha}$ from a uniform distribution $U[0; 1]$. As in the layout induction investigations, we experiment with several SQOOP splits, namely we try #rhs/lhs $\in \{1, 2, 18\}$. The results (reported in Table 2) show that Attention N2NMN fails dramatically on #rhs/lhs=1 but quickly catches up as soon as #rhs/lhs is increased to 2. Notably, 9 out of 10 runs on #rhs/lhs=2 result in almost perfect performance, and 1 run completely fails to generalize (26% error rate), resulting in a high $8.18\%$ variance of the mean

Table 1: Tree layout induction results for Stochastic N2NMNs using $\mathrm{Residual}$ and $\mathrm{Find}$ modules on 1 rhs/lhs and 18 rhs/lhs datasets. For each setting of $p_0(tree)$ we report results after 5 runs. $p_{200K}(tree)$ is the probability of using a tree layout after 200K training iterations.

| module | #rhs/lhs | $p_0(tree)$ | Test error rate (%) | Test loss | $p_{200K}(tree)$ |
|---|---|---|---|---|---|
| Residual | 1 | 0.1 | $31.89 \pm 0.75$ | $0.64 \pm 0.03$ | $0.08 \pm 0.01$ |
| | | 0.5 | $1.64 \pm 1.79$ | $0.27 \pm 0.04$ | $0.56 \pm 0.06$ |
| | | 0.9 | $0.16 \pm 0.11$ | $0.03 \pm 0.01$ | $0.96 \pm 0.00$ |
| | 18 | 0.1 | $3.99 \pm 5.33$ | $0.15 \pm 0.06$ | $0.59 \pm 0.34$ |
| | | 0.5 | $0.19 \pm 0.11$ | $0.06 \pm 0.02$ | $0.99 \pm 0.01$ |
| | | 0.9 | $0.12 \pm 0.12$ | $0.01 \pm 0.00$ | $1.00 \pm 0.00$ |
| Find | 1 | 0.1 | $47.54 \pm 0.95$ | $1.78 \pm 0.47$ | $0.00 \pm 0.00$ |
| | | 0.5 | $0.78 \pm 0.52$ | $0.05 \pm 0.04$ | $0.94 \pm 0.07$ |
| | | 0.9 | $0.41 \pm 0.07$ | $0.02 \pm 0.00$ | $1.00 \pm 0.00$ |
| | 18 | 0.1 | $5.11 \pm 1.19$ | $0.14 \pm 0.03$ | $0.02 \pm 0.04$ |
| | | 0.5 | $0.17 \pm 0.16$ | $0.01 \pm 0.01$ | $1.00 \pm 0.00$ |
| | | 0.9 | $0.11 \pm 0.03$ | $0.00 \pm 0.00$ | $1.00 \pm 0.00$ |

Table 2: Parameterization induction results for 1,2,18 rhs/lhs datasets for Attention N2NMN. The model does not generalize well in the difficult 1 rhs/lhs setting. Results for MAC are presented for comparison. Means and standard deviations were estimated based on at least 10 runs.

| Model | #rhs/lhs | Test error rate (%) | Test loss (%) |
|---|---|---|---|
| Attention N2NMN | 1 | $27.19 \pm 16.02$ | $1.22 \pm 0.71$ |
| Attention N2NMN | 2 | $2.82 \pm 8.18$ | $0.14 \pm 0.41$ |
| Attention N2NMN | 18 | $0.16 \pm 0.12$ | $0.00 \pm 0.00$ |
| MAC | 1 | $13.67 \pm 9.97$ | $0.41 \pm 0.32$ |
| MAC | 2 | $9.21 \pm 4.31$ | $0.28 \pm 0.15$ |
| MAC | 18 | $0.53 \pm 0.74$ | $0.01 \pm 0.02$ |

error rate. All 10 runs on the split with 18 rhs/lhs generalize flawlessly. Furthermore, we inspect the learned attention weights and find that for typical successful runs, module 3 focuses on the relation word, whereas modules 1 and 2 focus on different object words (see Figure 6) while still focusing on the relation word. To better understand the relationship between successful layout induction and generalization, we define an attention quality metric $\kappa = \min_{w \in \{X, Y\}} \max_{k \in 1,2} \alpha_{k,w}/(1 - \alpha_{k,R})$. Intuitively, $\kappa$ is large when for each word $w \in X, Y$ there is a module $i$ that focuses mostly on this word. The renormalization by $1/(1 - \alpha_{k,R})$ is necessary to factor out the amount of attention that modules 1 and 2 assign to the relation word. For the ground-truth parametrization that we use for NMN-Tree $\kappa$ takes a value of 1, and if both modules 1 and 2 focus on X, completely ignoring Y, $\kappa$ equals 0. The scatterplot of the test error rate versus $\kappa$ (Figure 5) shows that for #rhs/lhs=1 high generalization is strongly associated with higher $\kappa$, meaning that it is indeed necessary to have different modules strongly focusing on different object words in order to generalize in this most challenging setting. Interestingly, for #rhs/lhs=2 we see a lot of cases where N2NMN generalizes well despite attention being rather spurious ($\kappa \approx 0.6$).

In order to put Attention N2NMN results in context we compare them to those of MAC (see Table 2). Such a comparison can be of interest because both models perform attention over the question. For 1 rhs/lhs MAC seems to be better on average, but as we increase #rhs/lhs to 2 we note that Attention N2NMN succeeds in 9 out of 10 cases on the #rhs/lhs=2 split, much more often than 1 success out of 10 observed for MAC[2]. This result suggests that Attention N2NMNs retains some of the strong generalization potential of NMNs with hard-coded parametrization.

## 5 RELATED WORK

The notion of systematicity was originally introduced by (Fodor & Pylyshyn, 1988) as the property of human cognition whereby "the ability to entertain a given thought implies the ability to entertain thoughts with semantically related contents". They illustrate this with an example that no English

---

[2] If we judge a run successful when the error rate is lower than $\tau = 1\%$, these success rates are different with a p-value of 0.001 according to the Fisher exact test. Same holds for any other threshold $\tau \in [1\%; 5\%]$.

speaker can understand the phrase "John loves the girl" without being also able to understand the phrase "the girl loves John". The question of whether or not connectionist models of cognition can account for the systematicity phenomenon has been a subject of a long debate in cognitive science (Fodor & Pylyshyn, 1988; Smolensky, 1987; Marcus, 1998; 2003; Calvo & Colunga, 2003). Recent research has shown that lack of systematicity in the generalization is still a concern for the modern seq2seq models (Lake & Baroni, 2018; Bastings et al., 2018; Loula et al., 2018). Our findings about the weak systematic generalization of generic VQA models corroborate the aforementioned seq2seq results. We also go beyond merely stating negative generalization results and showcase the high systematicity potential of adding explicit modularity and structure to modern deep learning models.

Besides the theoretical appeal of systematicity, our study is inspired by highly related prior evidence that when trained on downstream language understanding tasks, neural networks often generalize poorly and latch on to dataset-specific regularities. Agrawal et al. (2016) report how neural models exploit biases in a VQA dataset, e.g. responding "snow" to the question "what covers the ground" regardless of the image because "snow" is the most common answer to this question. Gururangan et al. (2018) report that many successes in natural language entailment are actually due to exploiting statistical biases as opposed to solving entailment, and that state-of-the-art systems are much less performant when tested on unbiased data. Jia & Liang (2017) demonstrate that seemingly state-of-the-art reading comprehension system can be misled by simply appending an unrelated sentence that resembles the question to the document.

Using synthetic VQA datasets to study grounded language understanding is a recent trend started by the CLEVR dataset (Johnson et al., 2016). CLEVR images are 3D-rendered and CLEVR questions are longer and more complex than ours, but in the associated generalization split CLEVR-CoGenT the training and test distributions of images are different. In our design of SQOOP we aimed instead to minimize the difference between training and test images to make sure that we test a model's ability to interpret unknown combinations of known words. The ShapeWorld family of datasets by Kuhnle & Copestake (2017) is another synthetic VQA platform with a number of generalization tests, but none of them tests SQOOP-style generalization of relational reasoning to unseen object pairs. Most closely related to our work is the recent study of generalization to long-tail questions about rare objects done by Bingham et al. (2017). They do not, however, consider as many models as we do and do not study the question of whether the best-performing models can be made end-to-end.

The key paradigm that we test in our experiments is Neural Module Networks (NMN). Andreas et al. (2016) introduced NMNs as a modular, structured VQA model where a fixed number of hand-crafted neural modules (such as `Find`, or `Compare`) are chosen and composed together in a layout determined by the dependency parse of the question. Andreas et al. (2016) show that the modular structure allows answering questions that are longer than the training ones, a kind of generalization that is complementary to the one we study here. Hu et al. (2017) and Johnson et al. (2017) followed up by making NMNs end-to-end, removing the non-differentiable parser. Both Hu et al. (2017) and Johnson et al. (2017) reported that several thousands of ground-truth layouts are required to pretrain the layout predictor in order for their approaches to work. In a recent work, Hu et al. (2018) attempt to soften the layout decisions, but training their models end-to-end from scratch performed substantially lower than best models on the CLEVR task. Gupta & Lewis (2018) report successful layout induction on CLEVR for a carefully engineered heterogeneous NMN that takes a scene graph as opposed to a raw image as the input.

## 6 CONCLUSION AND DISCUSSION

We have conducted a rigorous investigation of an important form of systematic generalization required for grounded language understanding: the ability to reason about all possible pairs of objects despite being trained on a small subset of such pairs. Our results allow one to draw two important conclusions. For one, the intuitive appeal of modularity and structure in designing neural architectures for language understanding is now supported by our results, which show how a modular model consisting of general purpose residual blocks generalizes much better than a number of baselines, including architectures such as MAC, FiLM and RelNet that were designed specifically for visual reasoning. While this may seem unsurprising, to the best of our knowledge, the literature has lacked such a clear empirical evidence in favor of modular and structured networks before this work. Importantly, we have also shown how sensitive the high performance of the modular models is to the

layout of modules, and how a tree-like structure generalizes much stronger than a typical chain of layers.

Our second key conclusion is that coming up with an end-to-end and/or soft version of modular models may be not sufficient for strong generalization. In the very setting where strong generalization is required, end-to-end methods often converge to a different, less compositional solution (e.g. a chain layout or blurred attention). This can be observed especially clearly in our NMN layout and parametrization induction experiments on the #rhs/lhs=1 version of SQOOP, but notably, strong initialization sensitivity of layout induction remains an issue even on the #rhs/lhs=18 split. This conclusion is relevant in the view of recent work in the direction of making NMNs more end-to-end (Suarez et al., 2018; Hu et al., 2018; Hudson & Manning, 2018; Gupta & Lewis, 2018). Our findings suggest that merely replacing hard-coded components with learnable counterparts can be insufficient, and that research on regularizers or priors that steer the learning towards more systematic solutions can be required. That said, our parametrization induction results on the #rhs/lhs=2 split are encouraging, as they show that compared to generic models, a weaker nudge (in the form of a richer training signal or a prior) towards systematicity may suffice for end-to-end NMNs.

While our investigation has been performed on a synthetic dataset, we believe that it is the real-world language understanding where our findings may be most relevant. It is possible to construct a synthetic dataset that is bias-free and that can only be solved if the model has understood the entirety of the dataset's language. It is, on the contrary, much harder to collect real-world datasets that do not permit highly dataset-specific solutions, as numerous dataset analysis papers of recent years have shown (see Section 5 for a review). We believe that approaches that can generalize strongly from imperfect and biased data will likely be required, and our experiments can be seen as a simulation of such a scenario. We hope, therefore, that our findings will inform researchers working on language understanding and provide them with a useful intuition about what facilitates strong generalization and what is likely to inhibit it.

## ACKNOWLEDGEMENTS

We thank Maxime Chevalier-Boisvert, Yoshua Bengio and Jacob Andreas for useful discussions. This research was enabled in part by support provided by Compute Canada (www.computecanada.ca), NSERC, Canada Research Chairs and Microsoft Research. We also thank Nvidia for donating NVIDIA DGX-1 used for this research.

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

## A EXPERIMENT DETAILS

We trained all models by minimizing the cross entropy loss $\log p(y|x, q)$ on the training set, where $y \in \{\text{yes}, \text{no}\}$ is the correct answer, $x$ is the image, $q$ is the question. In all our experiments we used the Adam optimizer (Kingma & Ba, 2015) with hyperparameters $\alpha = 0.0001$, $\beta_1 = 0.9$, $\beta_2 = 0.999$, $\epsilon = 10^{-10}$. We continuously monitored validation set performance of all models during training, selected the best one and reported its performance on the test set. The number of training iterations for each model was selected in preliminary investigations based on our observations of how long it takes for different models to converge. This information, as well as other training details, can be found in Table 3.

Table 3: Training details for all models. The subsampling factor is the ratio between the original spatial dimensions of the input image and those of the representation produced by the stem. It is effectively equal to $2^k$, where $k$ is the number of 2x2 max-pooling operations in the stem.

| model | stem layers | subsampling factor | iterations | batch size |
|---|---|---|---|---|
| FiLM | 6 | 4 | 200000 | 64 |
| MAC | 6 | 4 | 100000 | 128 |
| Conv+LSTM | 6 | 4 | 200000 | 128 |
| RelNet | 8 | 8 | 500000 | 64 |
| NMN (Residual) | 6 | 4 | 50000 | 64 |
| NMN (Find) | 6 | 4 | 200000 | 64 |
| Stochastic NMN (Residual) | 6 | 4 | 200000 | 64 |
| Stochastic NMN (Find) | 6 | 4 | 200000 | 64 |
| Attention NMN (Find) | 6 | 4 | 50000 | 64 |

## B ADDITIONAL RESULTS FOR MAC MODEL

We performed an ablation study in which we varied the number of MAC units, the model dimensionality and the level of weight decay for the MAC model. The results can be found in Table 4.

Table 4: Results of an ablation study for MAC. The default model has 12 MAC units of dimensionality 128 and uses no weight decay. For each experiment we report means and standard deviations based on 5 repetitions.

| model | #rhs/lhs | train error rate (%) | test error rate (%) |
|---|---|---|---|
| default | 1 | $0.17 \pm 0.21$ | $13.67 \pm 9.97$ |
| 1 unit | 1 | $0.27 \pm 0.35$ | $28.67 \pm 1.91$ |
| 2 units | 1 | $0.23 \pm 0.13$ | $24.28 \pm 2.05$ |
| 3 units | 1 | $0.16 \pm 0.15$ | $26.47 \pm 1.12$ |
| 6 units | 1 | $0.18 \pm 0.17$ | $20.84 \pm 5.56$ |
| 24 units | 1 | $0.04 \pm 0.05$ | $9.11 \pm 7.67$ |
| dim. 64 | 1 | $0.27 \pm 0.33$ | $23.61 \pm 6.27$ |
| dim. 256 | 1 | $0.00 \pm 0.00$ | $4.62 \pm 5.07$ |
| dim. 512 | 1 | $0.02 \pm 0.04$ | $8.37 \pm 7.45$ |
| weight decay 0.00001 | 1 | $0.20 \pm 0.23$ | $19.21 \pm 9.27$ |
| weight decay 0.0001 | 1 | $1.00 \pm 0.54$ | $31.19 \pm 0.87$ |
| weight decay 0.001 | 1 | $40.55 \pm 1.35$ | $45.11 \pm 0.74$ |

We also perform qualitative investigations to understand the high variance in MAC's performance. In particular, we focus on control attention weights ($c$) for each run and aim to understand if runs that generalize have clear differences when compared to runs that failed. Interestingly, we observe that in successful runs each word $w \in X, Y$ has a unit that is strongly focused on it. To present our observations in quantitative terms, we plot attention quality $\kappa = \min_{w \in \{X,Y\}} \max_{k \in [1;12]} \alpha_{k,w}/(1 - \alpha_{k,R})$, where $\alpha$ are control scores vs accuracy in Figure 7 for each run (see Section 4.3.2 for an explanation of $\kappa$). We can clearly see a positive correlation between $\kappa$ and error rate, especially for low #rhs/lhs.

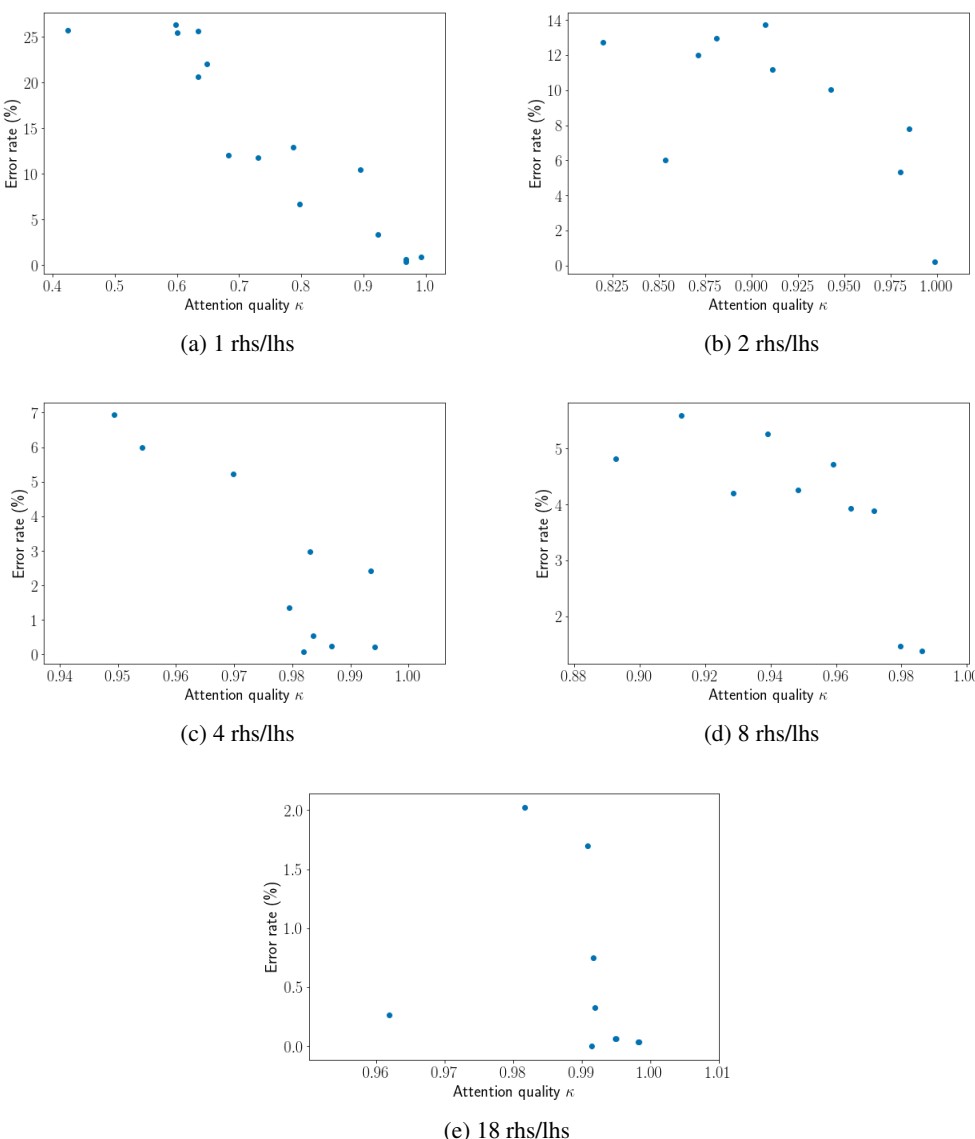

Figure 7: Model test accuracy vs $\kappa$ for the MAC model on different versions of SQOOP. All experiments are run 10 times with different random seeds. We can observe a clear correlation between $\kappa$ and error rate for 1, 2 and 4 rhs/lhs. Also note that perfect generalization is always associated with $\kappa$ close to 1.

Next, we experiment with a *hard-coded* variation of MAC. In this model, we use hard-coded control scores such that given a SQOOP question $X\,R\,Y$, the first half of all modules focuses on $X$ while the second half focuses on $Y$. The relationship between MAC and hardcoded MAC is similar to that between NMN-Tree and end-to-end NMN with parameterization induction. However, this model has not performed as well as the successful runs of MAC. We hypothesize that this could be due to the interactions between the control scores and the visual attention part of the model.

## C  INVESTIGATION OF CORRECT PREDICTIONS WITH SPURIOUS LAYOUTS

In Section 4.3.1 we observed that an NMN with the $\mathrm{Residual}$ module can answer test questions with a relative low error rate of $1.64 \pm 1.79\%$, despite being a mixture of a tree and a chain (see

results in Table 1, $p_0(tree) = 0.5$). Our explanation for this phenomenon is as follows: when connected in a tree, modules of such spurious models generalize well, and when connected as a chain they generalize poorly. The output distribution of the whole model is thus a mixture of the mostly correct $p(y|T = T_{tree}, x, q)$ and mostly random $p(y|T = T_{chain}, x, q)$. We verify our reasoning by explicitly evaluating test accuracies for $p(y|T = T_{tree}, x, q)$ and $p(y|T = T_{chain}, x, q)$, and find them to be around 99% and 60% respectively, confirming our hypothesis. As a result the predictions of the spurious models with $p(tree) \approx 0.5$ have lower confidence than those of sharp tree models, as indicated by the high log loss of $0.27 \pm 0.04$. We visualize the progress of structure induction for the Residual module with $p_0(tree) = 0.5$ in Figure 4 which shows how $p(tree)$ saturates to 1.0 for #rhs/lhs=18 and remains around 0.5 when #rhs/lhs=1.

## D  SQOOP PSEUDOCODE

---

**Algorithm 1** Pseudocode for creating SQOOP

---

1: $S \leftarrow \{A,B,C, \ldots, Z, 0,1,2,3, \ldots, 9\}$
2: $Rel \leftarrow \{\text{LEFT-OF, RIGHT-OF, ABOVE, BELOW}\}$        ▷ relations
3: **function** CREATESQOOP(k)
4:      $TrainQuestions \leftarrow []$
5:      $AllQuestions \leftarrow []$
6:      **for all** $X$ in $S$ **do**
7:          $AllRhs \leftarrow \text{RandomSample}(S \setminus \{X\}, k)$    ▷ sample without replacement from $S \setminus \{X\}$
8:          $AllQuestions \leftarrow \{X\} \times Rel \times (S \setminus \{X\}) \cup AllQuestions$
9:          **for all** $R, Y$ in $AllRhs \times Rel$ **do**
10:             $TrainQuestions \leftarrow (X, R, Y) \cup TrainQuestions$
11:          **end for**
12:      **end for**
13:      $TestQuestions \leftarrow AllQuestions \setminus TrainQuestions$
14:      **function** GENERATEEXAMPLE($X, R, Y$)
15:          $a \sim \{\text{Yes, No}\}$
16:          **if** $a = \text{Yes}$ **then**
17:             $I \leftarrow$ place $X$ and $Y$ objects so that $R$ holds          ▷ create the image
18:             $I \leftarrow$ sample 3 objects from $S$ and add to $I$
19:          **else**
20:             **repeat**
21:                 $X' \leftarrow$ Sample $X'$ from $S \setminus \{X\}$
22:                 $Y' \leftarrow$ Sample $Y'$ from $S \setminus \{Y\}$
23:                 $I \leftarrow$ place $X'$ and $Y$ objects so that $R$ holds         ▷ create the image
24:                 $I \leftarrow$ add $X$ and $Y'$ objects to $I$ so that $R$ holds
25:                 $I \leftarrow$ sample 1 more object from $S$ and add to $I$
26:             **until** $X$ and $Y$ are not in relation $R$ in I
27:          **end if**
28:          **return** $I, X, R, Y, a$
29:      **end function**
30:      $Train \leftarrow$ sample $\frac{10^6}{|TrainQuestions|}$ examples for each (X,R,Y) $\in TrainQuestions$ from GENERATEEXAMPLE($X, R, Y$)
31:      $Test \leftarrow$ sample 10 examples for each (X,R,Y) $\in TestQuestions$ from GENERATEEXAMPLE($X, R, Y$)
32: **end function**

---

