# OpenReview forum: "Systematic Generalization: What Is Required and Can It Be Learned?"
_ICLR.cc/2019/Conference_

### Official Review · AnonReviewer2 · 2018-11-02

**Rating:** 6
**Confidence:** 5

**Review:**

This paper presents a targeted empirical evaluation of generalization in models
for visual reasoning. The paper focuses on the specific problem of recognizing
(object, relation, object) triples in synthetic scenes featuring letters and
numbers, and evaluates models' ability to generalize to the full distribution of
such triples after observing a subset that is sparse in the third argument. It
is found that (1) NMNs with full layout supervision generalize better than other
state-of-the art visual reasoning models (FiLM, MAC, RelNet), but (2) without
supervised layouts, NMNs perform little better than chance, and without
supervised question attentions, NMNs perform better than the other models but
fail to achieve perfect generalization.

STRENGTHS
- thorough analysis with a good set of questions

WEAKNESSES
- some peculiar evaluation and presentation decisions
- introduces *yet another* synthetic visual reasoning dataset rather than
  reusing existing ones

I think this paper would have been stronger if it investigated a slightly
broader notion of generalization and had some additional modeling comparisons.
However, I found it interesting and think it successfully addresses the set of
questions it sets out to answer. If it is accepted, there are a few things that
can be done to improve the experiments.

MODELING AND EVALUATION

- Regarding the dataset: the proliferation of synthetic reasoning datasets is
  annoying because it makes it difficult to compare results without downloading
  and re-running a huge amount of code. (The authors have, to their credit, done
  so for this paper.) I think all the experiments here could have been performed
  successfully using either the CLEVR or ShapeWorld rendering engines: while the
  authors note that they require a "large number of different objects", this
  could have been handled by treating e.g. "red circle" and "red square" as
  distinct atomic primitives in questions---the fact that redness is a useful
  feature in both cases is no different from the fact that a horizontal stroke
  detector is useful for lots of letters.

- I don't understand the motivation behind holding out everything on the
  right-hand side. For models that can't tell that the two are symmetric, why
  not introduce sparsity everwhere---hold out some LHSs and relations?

- Table 1 test accuracies: arbitrarily reporting "best of 3" for some model /
  dataset pairs and "confidence interval of 5" for others is extremely
  unhelpful: it would be best to report (mean / max / stderr) for 5. Also, it's
  never stated which convidence interval is reported.

- Table 1 baselines: why not run Conv+LSTM and RelNet with easier #rhs/lhs data?

- How many MAC cells are used? This can have significant performance
  implications. I think if you used their code out of the box you'll wind up
  with way bigger structures than you need for this task.

- I'm not sure how faithful the `find` module used here is to the one in the
  literature, and one of the interesting claims in this work is that module
  implementation details matter! The various Hu papers use an attentional
  parameterization; the use of a ReLU and full convolution in Eq. 14 suggest
  that that one here can pass around more general feature maps. This is fine but
  the distinction should be made explicit, and it would be interesting to see
  additional comparisons to an NMN with purely attentional bottlenecks.

- Why do all the experiments after 4.3 use #rhs/lhs of 18? If it was 8 it would
  be possible to make more direct comparisons to the other baseline models.

- The comparison to MAC in 4.2 is unfair in the following sense: the NMN
  effectively gets supervised textual attentions if the right parameters are
  always plugged into the right models, while the MAC model has to figure out
  attentions from scratch. A different way of structuring things would be to
  give the MAC model supervised parameterizations in 4.2, and then move the
  current MAC experiment to 4.3 (since it's doing something analogous to
  "parameterization induction".

- The top-right number in Table 4---particularly the fact that it beats MAC and
  sequential NMNs under the same supervision condition---is one of the most
  interesting results in this paper. Most of the work on relaxing supervision
  for NMNs has focused on (1) inducing new question-specific discrete structures
  from scratch (N2NMN) or (2) finding fixed sequential structures that work well
  in general (SNMN and perhaps MAC). The result this paper suggests an
  alternative, which is finding good fixed tree-shaped structures but continuing
  to do soft parameterization like N2NMN.

- The "sharpness ratio" is not super easy to interpret---can't you just report
  something standard like entropy? Fig 4 is unnecessary---just report the means.

- One direction that isn't explored here is the use of Johnson- or Hu-style
  offline learning of a model to map from "sentences" to "logical forms". To the
  extent that NMNs with ground-truth logical forms get 100% accuracy, this turns
  the generalization problem studied here into a purely symbolic one of the kind
  studied in Lake & Baroni 18. Would be interesting to know whether this makes
  things harder (b/c no grounding signal) or easier (b/c seq2seq learning is
  easier.)

PRESENTATION

- Basically all of the tables in this paper are in the wrong place. Move them
  closer to the first metnion---otherwise they're confusing.

- It's conventional in this conference format to put all figure captions below
  the figures they describe. The mix of above and below here makes it hard to
  attach captions to figures.

- Some of the language about how novel the idea of studying generalization in
  these models is a bit strong. The CoGenT split of the CLEVR dataset is aimed
  at answering similar questions. The original Andreas et al CVPR paper (which btw
  appears to have 2 bib entries) also studied generalization to structurally
  novel inputs, and Hu et al. 17 notes that the latent-variable version of this
  model with no supervision is hard to train.

MISCELLANEOUS

- Last sentence before 4.4: "NMN-Chain" should be "NMN-Tree"?

- Recent paper with a better structure-induction technique:
  https://arxiv.org/abs/1808.09942. Worth citing (or comparing if you have
  time!)

---

> ### Author Response · Authors · 2018-11-17
> **Response to Reviewer 2**
>
> We thank Reviewer 2 (R2) for their excellent and thorough review and for raising several particularly interesting points about modeling and evaluation.
>
> While we do agree with the reviewer’s concerns that the proliferation of synthetic datasets may be counterproductive, we chose to create SQOOP instead of directly using existing datasets to keep things simple. R2 suggests that we could’ve defined new objects out of (color, shape) tuples. We believe though, that even if we used Blender (CLEVR) or ShapeWorld rendering to build a dataset for out studies, this would not make further experimentation any simpler, because even though the rendering would be the same, this would still constitute a new dataset. The entire code for generating SQOOP is merely 550 lines, and comes with an extremely simple set of command line arguments. This is to be contrasted with ~9500 lines of code in ShapeWorld codebase, which aims to be universally usable, and hence is highly convoluted. Furthermore, in order to help researchers avoid the burden of “downloading and re-running a huge amount of code”, we will release our codebase that contains implementations of all the models used in this study and comes with ready-to-use CLEVR and SQOOP bindings.
>
> We thank R2 for their thoughtful suggestion to consider splits other then the one with heldout right-hand sides (rhs). We fully agree that other options exist, for example a split where different lhsand rhs objects are used for each relation, and that investigating such options would be interesting. At the same time, we do not think that these extra experiments would radically change the conclusions, and we note that even in the current form our paper hits ICLR page limit. Our specific split was chosen based on the following considerations: we wanted to uniformly select a subset of object pairs for training set questions, in a way that guarantees that all objects word are seen at the training time. If we sampled a certain percentage of object pairs for training questions randomly, it could happen that certain words just never occur in the training set. Hence, we came up with the idea of having a fixed number of rhs objects for each lhs object. We note that this very split can also be seen as allowing a random (possibly zero) number of lhs objects for each rhs object, exhibiting sparsity on the lhs  like R2 suggested. We will better explain the considerations above in the upcoming paper revision.
>
> Apart from the above points of R2, we fully agree with their suggested changes and experiments and will incorporate almost all of these in the updated version of the paper.
>
> 1) We follow R2’s suggestions and improved the presentation in Table-1: we will report means and standard deviations for 5 runs for all our models.
> 2) CNN+LSTM and RelNet baselines are being re-run with higher #rhs/lhs.
> 3) We have run experiments with varying number of MAC cells (3,6,12,24) and found that using 12 cells performed best (and as well as using 24 cells). We believe that this has to do with lucky control score initializations. This, along with some new interesting qualitative investigations about the nature of control parameters that result in successful generalization, will be elaborated on in our updated manuscript.
> 4) In our initial experiments, we found that conceptually simpler homogenous NMNs (of the form proposed by Johnson et al.) are already sufficient to solve even the hardest version of SQOOP. Hence, we chose to focus our study on this, arguably, more generic approach, and we adapted the Find module from (Hu et al) to output a full feature map, instead of an attention map. We believe it is highly interesting to include such a model in comparison, as Residual and Find represent two very distinct paradigms of conditioning modules on their language inputs.  We agree that extra studies of NMNs with attention bottlenecks would be a interesting direction of the future work, but we also think that our paper is quite complete without this investigation and has enough interesting findings.
> 5) We will report performance of all baseline models on the #rhs/lhs=18 version of our dataset as well.
> 6) We also fully agree with R2’s excellent observation about the nature of supervision in MAC vs hard-coded parameter NMN models. We are now running MAC experiments with hardcoded control attention where the control scores are hard-coded such that some of the modules focus entirely on the LHS object and some focus entirely on the RHS object. This particular hard-coding strategy was a result of our qualitative understanding of successful learnt attention for MAC. We will elaborate on this in the paper.
> 7) We agree with R2’s comment that studying seq2seq learning in our setting would add an interesting new dimension to this work, and this is something we’ll consider for future work.
> 8) We also note R2’s feedback on strong language, presentation issues and a missing citation and will improve the paper in these aspects.

---

> ### Author Response · Authors · 2018-11-30
> **Kind request to respond for Reviewer 2**
>
> Dear Reviewer 2,
>
> Thank you once again for the thoughtful and thorough review that you wrote before the revision period. Our understanding of your review is that overall, you find the paper interesting and useful, but certain presentation and evaluation decisions, as well as the fact that we use a new dataset, did not allow you to recommend it stronger. Since then we have improved the paper a lot by incorporating a lot of your suggestions, including but not limited to reporting mean performance on at least 5 runs in all experiments, comparing MAC and Attention N2NMN, investigating different version of the MAC models. We have also argued extensively why we think our decision to build SQOOP from scratch, rather than rely on Blender or ShapeWorld’s rendering, will not have any negative consequences on our research field.
>
> A response from you on the updated version of our paper would be highly valuable to help us improve this work in the future. We would highly appreciate if you could take a look at the revised paper and let us know if you think it is still merely marginally above the acceptance threshold, or if perhaps you find that it already deserves a higher rating. We would be grateful even for a short response from you, highlighting what issues in the paper have not been addressed, or what arguments in our response are still are unconvincing.
>
> We are sincerely hoping to hear from you.

---

### Official Review · AnonReviewer1 · 2018-11-03
**Interesting observations but limited experiments; also doubtful how experiments and learning can be generalized to more complex tasks**

**Rating:** 6
**Confidence:** 3

**Review:**

Summary: The paper focuses on comparing the impact of explicit modularity and structure on systematic generalization by studying neural modular networks and “generic” models. The paper studies one instantiation of this systematic generalization for the setting of binary “yes” or “no” visual question answering task.  They introduce a new dataset called in which model has to answer questions that require spatial reasoning about pairs of randomly scattered letters and digits in the image. While the models are evaluated on all possible object pairs, they are trained on a smaller subset. They observe that NMNs generalize better than other neural models when an appropriate choice of layout and parametrization is made. They also show that current end-to-end approaches for inducing model layout or learning model parametrization fail to generalize better than generic models.

Pros:
- The conclusions of the paper regarding the generalization ability of neural modular networks is timely given the widespread interest in these class of algorithms.
- Additionally, they present interesting observations regarding how sensitive NMNs are to the layout of models. Experimental evidence (albeit on specific type of question) of this behaviour will be helpful for the community and hopefully motivate them to incorporate regularizers or priors that steer the learning towards better layouts.
- The authors provide a nice summary of all the models analyzed in Section 3.1 and Section 3.2.

Cons:
- While the results on SQOOP dataset are interesting, it would have been very exciting to see results on other synthetic datasets. Specifically, there are two datasets which are more complex and uses templated language to generate synthetic datasets similar to this paper:
    - CLEVR environment or a modification of that dataset to reflect the form of systematic the authors are studying in the paper.
    - Abstract Scenes VQA dataset introduced in“Yin and Yang: Balancing and Answering Binary Visual Questions” by Zhang and Goyal et al. They provide a balanced dataset in which there are a pairs of scenes for every question, such that the answer to the question is “yes” for one scene, and “no” for the other for the exact same question.
- Perhaps because the authors study a very specific kind of question, they limit their analysis to only three modules and two structures (tree & chain). However, in the most general setting NMN will form a DAG and it would have been interesting to see what form of DAGs generalize better than other.
- It is not clear to me how the analysis done in this paper will generalize to other more complex datasets where the network layout NMN might be more complex, the number of modules and type of modules might also be more. Because, the results are only shown on one dataset, it is harder to see how one might extend this work to other form of questions on slightly harder datasets.

Other Questions / Remarks:
- Given that the accuracy drop is very significant moving from NMN-Tree to NMN-Chain, is there an explanation for this drop?
- While the authors mention multiple times that #rhs/#lhs = 1 and 2 are more challenging than #rhs/#lhs=18, they do not sufficiently explain why this is the case anywhere in the paper.
- Small typo in the last line of section 4.3 on page 7. It should say: This is in stark contrast with “NMN-Tree” …..
- Small typo in the “Layout induction” paragraph, line 6 on Page 7:  … and for $p_0(tree) = 0.1$ and when we use the Find module

---

> ### Author Response · Authors · 2018-11-12
> **Response to Reviewer 1 (part 1 of 2)**
>
> We thank Reviewer 1 (R1) for their review and for asking interesting questions that helped us to understand where our paper may have been unclear. In our response below we will try our best to better explain our motivation for building and using SQOOP, as well as address R1’s other questions and concerns.
>
> A key concern that R1 expressed in their review is that we perform our study on the new SQOOP dataset, instead of using an available one (for example CLEVR or Abstract Scenes VQA). Though we appreciate the concern (it has spurred us to rethink and rephrase how we justify SQOOP) we still believe that the SQOOP dataset is the best choice for precisely testing our ideas. We kindly invite R1 to consider the following arguments in favor of doing so:
>
> The goal of our study was to perform a thorough investigation of systematic generalization of language understanding models. To that end, we wanted a setup that is as simple as possible, while still being challenging by testing the ability to extend the relational reasoning learned to unseen combinations of seen words. We therefore choose to focus on simplest relational questions of the form XRY, as they also allow us to factor out challenges of discrete optimization in choosing the right module layout (required for Stochastic N2NMN). The simplicity is also useful because most models get to 100% accuracy on the training set of SQOOP, which allowed us to put aside any remaining optimization challenges and just focus our study on systematic generalization.
> In contrast, we find that the popular CLEVR dataset does not satisfy our requirements and if we did modify it sufficiently, we believe that it would only differ from SQOOP in the actual rendering and would not affect our conclusions. Though visually more complex, CLEVR has only 3 object types: cylinder, sphere and cube. Therefore, it would only allow for 3x4x3=36 different XRY relational questions. This is arguably not enough to sufficiently represent real world situations, and would definitely hinder our experiments. Specifically, we would not be able to sufficiently vary the difficulty of our generalization challenge when allowing 1,2,4,8 or 18 possible right hand-side objects in the questions (we clarify why splits with lower #rhs/lhs are more difficult than those with higher #rhs/lhs later in this response). Hence, we did not find the original CLEVR readily appropriate for our study. We could, in theory, introduce new object types to CLEVR and rerender a new dataset in 3D using Blender (the renderer that was used to create CLEVR) with different lighting conditions and partial occlusions. Though enticing, we believe that such a 3D version of SQOOP would lead to exactly same conclusions, because the vision required to recognize the objects in the scene would still be rather trivial.
> The Ying and Yang dataset is clearly a valuable resource (and we thank the reviewer for the pointer), but we do not think it is readily suitable for the kind of study that we aim to perform. The dataset, to the best of our understanding, uses crowd-sourced questions (as the questions are taken from Abstract VQA dataset, whose captions were entered by a human, according to the original VQA paper https://arxiv.org/pdf/1505.00468v6.pdf). Using crowd-sourced questions would not allow us to control our experiments at the level of precision that we wanted to achieve (e.g. we would not know the ground-truth layouts, it would be harder to construct splits of varying difficulty, etc.). As well, Abstract VQA contains only 50k scenes, and from our experience with SQOOP we know that this number would be not sufficient to rule out overfitting to training images as a factor.
>
> We thank R1 for their constructive suggestion to consider NMNs that form a DAG.  We are currently investigating a chain-structured NMN with shortcuts from the output of the stem to each of the modules, and we will soon report these additional results in the upcoming revision of the paper. We hope that these results, combined with further qualitative investigations we are conducting, will answer the legitimate question of R1 as to why Chain-NMN performs so much worse than Tree-NMN.
>
> We acknowledge that the text of the paper can be improved to explain better why splits with lower #rhs/lhs are generally harder than those with higher #rhs/lhs, and we thank R1 for pointing this out. Our reasoning is that lower #rhs/lhs are harder because the training admits more spurious solutions in them. In such spurious regimes models adapt to the specific lhs-rhs combinations from the training and can not generalize to unseen lhs-rhs combinations (i.e. generalizing from questions about “A” in relation with “B” to “A” in relation to “D” (as in #rhs/lhs=1) is more difficult than generalizing from questions about “A” in relation to “B” and “C” to the same “A” in relation to “D” (as in #rhs/lhs=2). We will update the paper to be more explicit in explaining these considerations.

---

> ### Author Response · Authors · 2018-11-12
> **Response to Reviewer 1 (part 2 of 2)**
>
> We would like to conclude our response by replying to the higher-level concern of R1 that the findings of our study may not “generalize to other more complex datasets where the network layout NMN might be more complex, the number of modules and type of modules might also be more”. While we fully agree that more complex datasets with more complex questions would bring new challenges, these are ones we purposely put aside (such as the general unavailability of ground-truth layouts for vanilla NMN, the need to consider an exponentially large set of possible layouts for Stochastic N2NMN, etc.) We believe that it is highly valuable for the research community to know what happens in the simple ideal case of SQOOP, where we can precisely test our specific generalization criterion. This knowledge (e.g. the superiority of trees to chains, the sensitivity of layout induction to initialization, the emergence of spurious parameterization in end-to-end learning), will guide researchers in choosing, designing and troubleshooting their models, as they now know what to expect modulo the optimization challenges that they may face. The field of language understanding with deep learning is not easily amenable to mathematical theoretical investigations and, with that in mind, rigorous minimalistic studies like ours are arguably very important. To some extent, they play the role of the former: they inform researcher intuition and lay a solid foundation for scientific dialogue. We purposely traded breadth for depth in our investigations, and we will go even deeper in the additional experiments that the upcoming revision will contain. We believe that the total of our results makes a complete conference paper. All that said, we would welcome specific suggestions of additional experiments that we could carry out in order to better validate our claims.
>
> We hope that this response has clarified to R1 what our paper was insufficiently clear about. A new revision with additional experiments and fixed typos will soon be uploaded to OpenReview, and we hope that R1 takes this response and the changes that we will make to the paper into account.

---

> > ### Comment · AnonReviewer1 · 2018-11-29
> > **Thanks for the detailed rebuttal, makes it a stronger paper**
> >
> > Thank you for your detailed responses and updates to the paper. I do think the updates made in the paper makes it  more clear and above acceptance threshold. I am convinced that it successfully analyzes an interesting set of questions and carefully studies this in a specific (albeit slightly narrow) notion of generalization.
> >
> > Therefore, I am updating the rating to above acceptance threshold.

---

### Official Review · AnonReviewer3 · 2018-11-04
**Interesting, but please add more experiments like this**

**Rating:** 4
**Confidence:** 4

**Review:**

The paper explores how well different visual reasoning models can learn systematic generalization on a simple binary task. They create a simple synthetic dataset, involving asking if particular types of objects are in a spatial relation to others. To test generalization, they lower the ratio of observed  combinations of objects in the training data. The authors show the result that tree structured neural module networks generalize very well, but other strong visual reasoning approaches do not. They also explore whether appropriate structures can be learned. I think this is a very interesting area to explore, and the paper is clearly written and presented.

As the authors admit, the main result is not especially surprising. I think everyone agrees that we can design models that show particular kinds of generalization by carefully building inductive bias into the architecture, and that it's easy to make these work on the right toy data. However, on less restricted data, more general architectures seem to show better generalization (even if it is not systematic). What I really want this paper to explore is when and why this happens. Even on synthetic data, when do or don't we see generalization (systematic or otherwise) from NMNs/MAC/FiLM? MAC in particular seems to have an inductive bias that might make some forms of systematic generalization possible. It might be the case that their version of NMN can only really do well on this specific task, which would be less interesting.

All the models show very high training accuracy, even if they do not show systematic generalization. That suggests that from the point of view of training, there are many equally good solutions, which suggests a number of interesting questions. If you did large numbers of training runs, would the models occasionally find the right solution? Could you somehow test for if a given trained model will show systematic generalization? Is there any way to help the models find the "right" (or better) solutions - e.g. adding regularization, or changing the model size?

Overall, I do think the paper has makes a contribution in experimentally showing a setting where tree-structured NMNs can show better systematic generalization than other visual reasoning approaches. However, I feel like the main result is a bit too predictable, and for acceptance I'd like to see a much more detailed exploration of the questions around systematic generalization.

---

> ### Author Response · Authors · 2018-11-12
> **Response to Reviewer 3**
>
> We thank Reviewer 3 (R3) for their review and for clearly articulating their concerns regarding the paper. In our response below, we will clarify the design and results of our experiments as well as argue why we believe that these results should be of interest and are not, indeed, that predictable.
>
> R3 asked why training performance of many models is 100% when they do not generalize and suggested us to perform a large number of training runs to see if occasionally the right solution is found. First, we agree that from the point of view of training there are many equally good solutions, and in fact, this is the main and the only challenge of SQOOP. We designed the task with the goal of testing which models are more likely to converge to the right solution, with which they can handle all possible combinations of objects, despite being trained only on a small subset of objects. We argued extensively in the introduction that such an ability to find the systematic solution despite other alternatives being available is highly desirable for language understanding approaches. We fully agree with R3 that in investigations of whether or not a particular model converges to the right solution repeating every experiment several times is absolutely necessary, and we would like to emphasize that we did repeat each experiment 3, 5, or 10 times (see “details” in Table 1 and the paragraph “Parametrization Induction” on page 8). In most cases we saw a consistent success or consistent failure, one exception being the parametrization induction results, where 4 out of 10 runs were successful (see Table 4, row 1 for the mean and the confidence interval). We hope that 3 takes this fact into account, and we will furthermore improve on the current level of rigor in the upcoming revision by repeating each experiment at least 5 times.
>
> We are not sure if we fully understand the question “Could you somehow test for if a given trained model will show systematic generalization?” that R3 asked. We test the systematic generalization of a model by evaluating it on all SQOOP questions that were not present in the training set. We hope that this answers the question of R3 and we would be happy to engage in a further discussion regarding this and make edits to the paper if necessary.
>
> We thank R3 for the suggestion to investigate the influence of model size and regularization on systematic generalization. It is indeed a very appropriate question in the  context of our study, however, we note that there exists a wide variety of regularization methods and trying them all (and all their combinations) would be infeasible. In the upcoming update of the paper we will report results of an on-going ablation study for the MAC model, in which we vary the module size, the number of modules and experiment with weight decay. We would welcome any other specific experiment requests R3 may have.
>
> Finally, we would like to discuss the significance of our investigation and its results. While we agree that the results that we report may not shock the reader (although perhaps hindsight bias plays a role in what people find surprising or not after reading an article) we find them highly interesting and not at all easily predictable. Reading prior work on visual reasoning may lead a researcher to conclude, roughly speaking, that NMNs are a lost cause, since a variety of generic models perform comparably or better. In contrast, our rigorous investigation highlights their strong generalization capabilities and relates them to the specific design of NMNs. Notably, chain-structured NMNs were used in the literature prior to this work (e.g. in the model of Jonshon et al multiple filter_...[...] modules are often chained), so the fact that tree-structured NMNs show much stronger generalization was not obvious prior to this investigation and should be of a high interest to the research community. Last but not least, an important part of our investigation (which the review does not discuss) is the systematic generalization analysis of popular end-to-end NMN versions, that shows how making NMNs more end-to-end makes them more susceptible to finding spurious solutions. As we argued in our conclusion, these findings should be of a highest importance to researchers working on end-to-end NMNs, which is a very popular research direction nowadays.
>
> We conclude our response by announcing that an updated version of the paper, that among others incorporates valuable suggestions by R3, will soon be uploaded to OpenReview. We are currently performing a lot of additional experiments, the results of which will make our investigation even more rigorous and complete. We sincerely hope that R3 takes into account the arguments we have made here and the new results that we will publish soon and reevaluates our paper more positively.

---

> > ### Comment · AnonReviewer3 · 2018-12-01
> > **Thanks for the response!**
> >
> > Thanks for the response, and sorry for the slow reply!
> >
> > After reading the response and revised paper, I'm leaving my review score unchanged, because I think my main concerns still stand. I didn't find the results surprising, and I don't see evidence that these results would generalize to more complex tasks. I think if the paper is only reporting experiments on a toy task, it would need to uncover something really interesting. That said, I would encourage the authors to keep working on this exciting topic.
> >
> > > Reading prior work on visual reasoning may lead a researcher to conclude, roughly speaking, that NMNs are a lost cause, since a variety of generic models perform comparably or better. In contrast, our rigorous investigation highlights their strong generalization capabilities and relates them to the specific design of NMNs.
> >
> > I don't find this argument convincing. For example, we could easily design a rule-based system that would show very strong generalization abilities on your task. However, that would not persuade me that it rule-based methods are not a lost cause for visual reasoning. I would really like to see some evidence that your results would generalize to more realistic tasks.
> >
> > > Notably, chain-structured NMNs were used in the literature prior to this work (e.g. in the model of Jonshon et al multiple filter_...[...] modules are often chained), so the fact that tree-structured NMNs show much stronger generalization was not obvious prior to this investigation and should be of a high interest to the research community.
> > As mentioned by another reviewer, “Neural Compositional Denotational Semantics for Question Answering” shows systematic generalization with tree structured NMNs, and goes much further with structure learning. I think you should at least explain how your results relate to this paper.
> >
> > > We are not sure if we fully understand the question “Could you somehow test for if a given trained model will show systematic generalization?” that R3 asked.
> > Sorry that this was unclear. I was wondering if you could test for this property without actually running on test data (maybe it converges faster, or the norm of the weights is lower; I have no idea). Knowing that might help us to regularize models properly during training.
> >
> > > All these experiments are repeated at least 5 times each, like you suggested in your review, although it’s worth noting that results the original version of the paper also reported results after  multiple runs.
> >
> > By "large numbers of runs", I was thinking more like thousands than five (I don't know if that is computational practical). The question I  was curious about is whether these models will ever find the right solution, or perhaps if they even have an inductive bias against finding it. This would be very helpful to know.

---

> ### Author Response · Authors · 2018-11-30
> **Kind request to respond for Reviewer 3**
>
> Dear Reviewer 3,
>
> We thank you again for your informative review that you wrote before the revision period. In our response and the revised version of the paper we tried our best to address your concerns. We would highly appreciate to get some feedback from you regarding the changes that we have made and the arguments that we have presented. In particular, we report that NMN-Chains (with a lot of inductive bias built-in and also used in prior work such as Johnson et al. 2017) generalize poorly compared to even generic modules, and that layout/parameterization induction often fails to converge to the correct solution. We believe both these findings are quite surprising. We also report new experiments with the MAC model, including a hyperparameter search, a comparison against end-to-end NMNs, and a qualitative exploration of the failure modes of this model. All these experiments are repeated at least 5 times each, like you suggested in your review, although it’s worth noting that results the original version of the paper also reported results after  multiple runs.
>
> We would highly appreciate a response on our newest revision and suggestions on how it could be improved. If you still think that paper is uninteresting or not well executed, could you then suggest what specifically it is lacking?
>
> We are sincerely hoping to hear from you.

---

### Author Response · Authors · 2018-11-26
**a greatly improved revision has been uploaded**

We are happy to present a new, substantially improved revision of the paper. We have polished our experimental setup (see details in the end of the message), performed many additional experiments as requested by the reviewers and improved presentation of the results.

Most important changes in the revision include:

1) We report means and standard deviation for at least 5 (and at least 10 in some comparisons due to variance in performance) runs of each of the models. We switched to reporting error rates instead of accuracies in all tables in order to make our results easier to understand.
2) Performance of MAC baseline has somewhat improved, compared to what we reported in the original submission, but this model is still far from solving SQOOP for #rhs/lhs of 1, 2, 4, 8, and it fails sometimes even on #rhs/lhs=18. We performed an ablation study of MAC as requested by R2  and R3, in which we varied the number of hidden units, the number of modules and the level of weight decay (see Appendix B). Results for all hyperparameters settings that we tried are still hopelessly far from systematic generalization of the kind exhibited by NMN-Tree, although on average MAC models with 256 hidden units performed somewhat better (barely statistically significantly) than the default version with 128 hidden units that we used in our experiments. We also now report qualitative analysis of rare (3 out of 15) cases when MAC does generalize, showing that this is likely to be due to a lucky initialization.
3) As suggested by R1, we added a DAG-like NMN-Chain-Shortcut model to the comparison. We found that its generalization performance is in between those of NMN-Chain and NMN-Tree and is in general quite similar to the performance of generic models.
4) We present additional results for NMN-Chain, showing that it does not generalize even when #rhs/lhs=18! We find this drastic lack of generalization highly surprising and not at all easily predictable without performing our study.
5) We performed an analysis of the responses produced by an NMN-Chain model to answer R1’s question as to why it performs so much worse than NMN-Tree. Our analysis has shown that there is almost no agreement in test set responses of several NMN-Chain models, allowing us to conclude that NMN-Chain essentially predicts randomly on the test set.
6) The results of layout induction experiments have somewhat improved, without major changes to the conclusions.
7) Perhaps the most significant changes have occured in our parametrization induction results. We found that Attention N2NMN may generalize quite well (9 times out of 10) even for #rhs/lhs=2, and most unexpectedly, even when attention weights are not very sharp. The results on #rhs/lhs=1 have remained the same. Our new results suggest that Attention N2NMN lends itself to systematic generalization more than MAC, supporting the hypothesis expressed by R2.

Other changes include:
1) We cite “Neural Compositional Denotational Semantics for Question Answering”, as suggested by R2.
2) We state explicitly in the text that our Find module outputs feature maps instead of attention maps, somewhat differently from the original Find modules from Hu et al.
3) Appendix A with training details has been added.
4) Appendix B with some qualitative analysis about why some MAC runs generalized successfully and others failed. We also report an attempt to hard-code control scores (as requested by R2) in MAC but that did not improve performance.
5) We explain the motivation for the dataset generation procedure more clearly in Section 2, and also follow a suggestion by R3 and explain better why lower rhs/lhs is harder for generalization.

We thank all reviewers for their valuable suggestions that allowed us to greatly improve the paper. We believe that the revised paper should be of a high interest for anyone working on language understanding, and we sincerely hope that reviewers will consider revisiting their evaluations.

P. S. The changes in the results were caused by the following improvements in the experimental setup:
1) We disabled the weight decay of 0.00001 that was the default in the codebase on top of which we start our project. This change allows for rare convergence to systematic solutions on the #rhs/lhs=1 split for MAC (3/15 runs). .
2) We found that the publicly available codebase for the FiLM model had redundant biases before batch normalization, and removing this redundancy has stabilised training on NMNs with Find module, including Attention NMNs.
3) In our preliminary experiments we set the learning rate for structural parameters to be higher than the one used for regular weights (0.001 vs 0.0001). To simplify our setup, we reran all experiments with the same learning rate for all parameters.

---

### Meta-Review · Area_Chair1 · 2018-12-14

**Confidence:** 4
**Recommendation:** Accept (Poster)

**Metareview:**

This paper generated a lot of discussion. Paper presents an empirical evaluation of generalization in models for visual reasoning. All reviewers generally agree that it presents a thorough evaluation with a good set of questions. The only remaining concerns of R3 (the sole negative vote) were lack of surprise in findings and lingering questions of whether these results generalize to realistic settings. The former suffers from hindsight bias and tends to be an unreliable indicator of the impact of a paper. The latter is an open question and should be worked on, but in the opinion of the AC, does not preclude publication of this manuscript. These experiments are well done and deserve to be published. If the findings don't generalize to more complex settings, we will let the noisy process of science correct our understanding in the future.